# Targeting Immunosuppressive Tumor-Associated Macrophages Using Innate T Cells for Enhanced Antitumor Reactivity

**DOI:** 10.3390/cancers14112749

**Published:** 2022-06-01

**Authors:** Yan-Ruide Li, James Brown, Yanqi Yu, Derek Lee, Kuangyi Zhou, Zachary Spencer Dunn, Ryan Hon, Matthew Wilson, Adam Kramer, Yichen Zhu, Ying Fang, Lili Yang

**Affiliations:** 1Department of Microbiology, Immunology & Molecular Genetics, University of California, Los Angeles, CA 90095, USA; charlie.li@ucla.edu (Y.-R.L.); brownjimw0@gmail.com (J.B.); yu63@g.ucla.edu (Y.Y.); ylee932@ucla.edu (D.L.); kuangyizhou@g.ucla.edu (K.Z.); zacharsd@usc.edu (Z.S.D.); ryanhonchenghao@gmail.com (R.H.); mwilson193@g.ucla.edu (M.W.); akramer931@g.ucla.edu (A.K.); yichenzhujohn@g.ucla.edu (Y.Z.); yingfang1913@g.ucla.edu (Y.F.); 2Mork Family Department of Chemical Engineering and Materials Science, University of Southern California, Los Angeles, CA 90089, USA; 3Eli and Edythe Broad Center of Regenerative Medicine and Stem Cell Research, University of California, Los Angeles, CA 90095, USA; 4Jonsson Comprehensive Cancer Center, David Geffen School of Medicine, University of California, Los Angeles, CA 90095, USA; 5Molecular Biology Institute, University of California, Los Angeles, CA 90095, USA

**Keywords:** tumor-associated macrophage, innate T cell, cancer immunotherapy, mucosal-associated invariant T (MAIT) cell, invariant natural killer T (iNKT) cell, gamma delta T (γδT) cells, tumor microenvironment (TME)

## Abstract

**Simple Summary:**

This study seeks to evaluate innate T cells for antitumor therapies that can target both tumor cells and immunosuppressive tumor-associated macrophages (TAMs). With their innate immune-like nature, mucosal-associated invariant T (MAIT) cells, invariant natural killer T (iNKT) cells, and gamma delta T (γδT) cells do not demonstrate the harmful graft-versus-host effects associated with many conventional αβ T cell trials, and could be engineered to target both tumor cells and anti-inflammatory TAMs. The success of these trials could suggest potent new avenues for the field of cell-based cancer immunotherapy.

**Abstract:**

The field of T cell-based and chimeric antigen receptor (CAR)-engineered T (CAR-T) cell-based antitumor immunotherapy has seen substantial developments in the past decade; however, considerable issues, such as graft-versus-host disease (GvHD) and tumor-associated immunosuppression, have proven to be substantial roadblocks to widespread adoption and implementation. Recent developments in innate immune cell-based CAR therapy have opened several doors for the expansion of this therapy, especially as it relates to allogeneic cell sources and solid tumor infiltration. This study establishes in vitro killing assays to examine the TAM-targeting efficacy of MAIT, iNKT, and γδT cells. This study also assesses the antitumor ability of CAR-engineered innate T cells, evaluating their potential adoption for clinical therapies. The in vitro trials presented in this study demonstrate the considerable TAM-killing abilities of all three innate T cell types, and confirm the enhanced antitumor abilities of CAR-engineered innate T cells. The tumor- and TAM-targeting capacity of these innate T cells suggest their potential for antitumor therapy that supplements cytotoxicity with remediation of tumor microenvironment (TME)-immunosuppression.

## 1. Introduction

A novel anticancer therapy that has seen considerable developments in the last decade is chimeric antigen receptor (CAR)-engineered T (CAR-T) cell therapy, which grafts a tumor antigen-specific, immunostimulatory ligand-independent receptor onto effector T cells [1]. These therapies show promise, but there are several roadblocks that have prevented their widespread adoption as first-response treatments. One current issue with CAR-T therapy is that CAR-T cells are susceptible to many of the same shortcomings as native T cells, such as antigen escape and exhaustion [1,2]. Additionally, the current state of CAR-T therapy requires autologous T cells to minimize the graft-versus-host (GvH) response that is typical of allogeneic CAR-T therapy, making it incredibly expensive and patient-specific [3]. These features limit the potential of conventional CAR-T cells in becoming an “off-the-shelf” version of the treatment.

Due to the highly conserved nature of the innate immune system, CAR-engineered innate T cells have demonstrated a considerably lower incidence of GvH response, positioning them as ideal candidates for antitumor CAR therapy [4,5]. Innate T cells are a group of T lymphocytes that display innate cell-like features, and are rapidly responsive to foreign pathogen signals [6]. Similar to conventional αβ T cells, innate T cells also undergo TCR rearrangement and thymic selection. Three major populations of innate T cells are recognized, namely, mucosal-associated invariant T (MAIT) cells, invariant natural killer T (iNKT) cells, and gamma delta T (γδT) cells [7,8]. MAIT cells are antibacterial immune cells that target invasive bacterial pathogens through the MAIT TCR recognition of unstable bacterial pyrimidines presented by the MHC class I-related gene protein (MR1) [9,10]. This allows them to have an MHC-independent response against bacterial invaders [10]. In a similar but distinctively unique fashion, iNKT cells recognize the presentation of both self and non-self lipid antigens by CD1d on antigen-presenting cells (APCs) [11]. These cells combine innate and adaptive immune characteristics, making them a sort of “hybrid” immune cell that has the adaptive advantages of typical αβ T cells and the rapid response of innate immunity [10,12]. The third innate T cells are γδT cells, which have a γδ TCR instead of the conventional αβ TCR, making them MHC-independent and allowing for an innate immune response to tumor-associated antigens and non-peptidic antigens [13]. γδT cells are also involved in immunoregulatory signaling, and can stimulate recruitment of other immune cell types [14].

Due to their perceived avoidance of GvH disease, these innate T cells are attractive CAR therapy candidates; however, the immunosuppressive TME remains a significant roadblock to solid tumor efficacy [15]. Tumor-associated macrophages (TAMs) constitute a heterogeneous and immunosuppressive cell population in the TME that accounts for up to 50% of solid tumors [16,17]. Moreover, TAMs support disease progression and induce tumor cell resistance to therapies by providing malignant cells with structural and trophic subsistence [18]. TAM-mediated suppression of immune cell antitumor reactivity is regarded as a major hurdle for many immunotherapies, including immune checkpoint blockade and adoptive T/CAR-T cell therapies [19]. For instance, immunosuppressive TAMs prevent the infiltration, and promote the exhaustion, of effector T cells through the production of anti-inflammatory cytokines (e.g., IL-10) and the expression of Programmed Cell Death Ligand 1 (PD-L1) [20]. Therefore, targeting and inhibiting TAMs has become a subject of considerable interest in current therapies [21,22].

Previous studies have identified that innate T cells, such as iNKT cells, can remediate immunosuppression through the targeting of anti-inflammatory TAMs [23]. Therefore, this study seeks to examine the efficacy of MAIT, iNKT, and γδT cells in targeting alternatively activated macrophages in vitro, and to assess their application in CAR therapies. The success of these trials could establish these cell types as effective modulators of the TME and lay the groundwork for the development of multipronged CAR therapies employing innate immune cells.

## 2. Materials and Methods

### 2.1. Cell Lines

Human melanoma cell line A375, human multiple myeloma cell line MM.1S, and human ovarian cancer cell line OVCAR3 were purchased from the ATCC (American Type Culture Collections, VA, USA). A375-FG, MM.1S-FG, and OVCAR3-FG were established by transduction of parental cell lines with Lenti/Fluc-EGFP lentivirus, as previously described [4,24]. A375-CD1d-FG and MM.1S-CD1d-FG were established by co-transduction of parental cell lines with Lenti/Fluc-EGFP and Lenti/CD1d lentiviruses, as previously described [24]. All these cells were maintained in RPMI 1640 medium supplemented with 10% (*v*/*v*) fetal bovine serum (FBS) and 1% (*v*/*v*) Penicillin–Streptomycin–Glutamine (P/S/G).

### 2.2. Antibodies and Flow Cytometry

Antibodies to human CD3 (Clone HIT3a, Pacific blue or PE-conjugated, 1:500), CD4 (Clone OKT4, Pacific blue or FITC-conjugated, 1:400), CD8 (Clone SK1, FITC, PE or APC-Cy7-conjugated, 1:400), TCRαβ (Clone I26, PE-Cy7 or Pacific blue-conjugated, 1:50), TCR Vα7.2 (Clone 3C10, APC or APC-Cy7-conjugated, 1:50), TCR Vδ2 (Clone B6, PerCP-conjugated, 1:100), MR1 (Clone 26.5, PE-conjugated, 1:50), CD1d (Clone 51.1, APC-conjugated, 1:50), CD11b (Clone ICRF44, FITC-conjugated, 1:10,000), CD14 (Clone HCD14, Pacific blue-conjugated, 1:1000), CD163 (Clone GHI/61, APC-Cy7-conjugated, 1:500), CD206 (Clone 15-2, APC-conjugated, 1:500), and CD25 (Clone BC96, PE-conjugated, 1:2000) were purchased from BioLegend. Antibody to human TCR Vα24-Jβ18 (Clone 6B11, PE-conjugated, 1:10) was purchased from BD Biosciences. Antibodies to human CD277 (Clone 849203, PE-conjugated, 1:50) and MSLN (Clone 420411, PE-conjugated, 1:20) were purchased from R&D Systems. Human Fc Receptor Blocking Solution (TruStain FcX) was purchased from BioLegend (San Diego, CA, USA), and Mouse Fc Block (anti-mouse CD16/32) was purchased from BD Biosciences (San Jose, CA, USA). Fixable Viability Dye e506 was purchased from Affymetrix eBioscience (San Diego, CA, USA).

Flow cytometry stains were performed in phosphate-buffered saline (PBS) for 15 min at 4 °C. The cell samples were stained with Fixable Viability Dye e506 mixed with Mouse Fc Block or Human Fc Receptor Blocking Solution prior to antibody staining. Antibody staining was performed according to the manufacturer’s instructions. Stained cell samples were analyzed using a MACSQuant Analyzer 10 flow cytometer (Miltenyi Biotec, Auburn, CA, USA) and FlowJo 9 software (FlowJo).

### 2.3. Enzyme-Linked Immunosorbent Assay (ELISA)

Cell culture supernatants were collected for cytokine ELISA analysis following a standard protocol from the BD Biosciences. The coating and biotinylated antibodies for the detection of human IFN-γ (coating antibody, catalog no. 551221; biotinylated detection antibody, catalog no. 554550) were purchased from BD Biosciences. The streptavidin–horseradish peroxidase (HRP) conjugate (catalog no. 18410051) was purchased from Invitrogen. Human IFN-γ (catalog no. 29-8319-65) standard was purchased from eBioscience. The 3,3′,5,5′-tetramethylbenzidine (TMB; catalog no. 51200048) substrate was purchased from Kirkegaard & Perry Laboratories (KPL, Gaithersburg, MD, USA). The absorbance at 450 nm was measured using an Infinite M1000 microplate reader (Tecan, Morrisville, NC, USA).

### 2.4. Generation of Human M2-Polarized Macrophage

Healthy donor peripheral blood mononuclear cells (PBMCs) were obtained from the CFAR Gene and Cellular Therapy Core Laboratory at UCLA, without identification information, under federal and state regulations. PBMCs were cultured in serum-free RPMI 1640 media (Corning cellgro, Manassas, VA, USA, #10-040-CV) at 1 × 10^7^ cells/mL cell density. Subsequently, 10–15 mL of the PBMC suspension was seeded into a 10 cm dish and incubated for 1–2 h in a humidified 37 °C, 5% CO_2_ incubator. Next, medium containing non-adherent cells was discarded and the dishes were washed twice using PBS. The adherent monocytes were cultured in RPMI 1640 medium supplemented with FBS (10% *v/v*), P/S/G (1% *v*/*v*), MEM NEAA (1% *v*/*v*), HEPES (10 mM), sodium pyruvate (1 mM), β-ME (50 mM), and Normocin (100 mg/mL) (denoted as C10 medium), and human M-CSF (10 ng/mL; PeproTech, #300-25) for 6 days to generate monocyte-derived macrophages (MDMs). At day 6, the generated MDMs were dissociated by 0.25% Trypsin/EDTA (Gibco, Waltham, MA, USA, #25200-056), collected, and reseeded in a 6-well plate in C10 medium at 0.5–1 × 10^6^ cells/mL cell density for 48 h in the presence of recombinant human IL-4 (10 ng/mL; PeproTech, #214-14) and human IL-13 (10 ng/mL; PeproTech, Rocky Hill, NJ, USA, #214-13) to induce MDM polarization. Polarized MDMs were then collected and used for flow cytometry or for setting up in vitro mixed culture experiments.

### 2.5. Generation of Human Mucosal-Associated Invariant T (MAIT), Invariant Natural Killer T (iNKT), Gamma Delta T (γδT), and Conventional Alpha Beta T (αβT) Cells

Healthy donor PBMCs obtained from the UCLA/CFAR Virology Core Laboratory were used to generate the MAIT, iNKT, γδT, and αβT cells.

To generate MAIT cells, PBMCs were MACS-sorted via PE-conjugated anti-TCR Vα7.2 antibody (BioLegend, San Diego, CA, USA) and anti-PE MicroBead (Miltenyi Biotec, Bergisch Gladbach, Germany) labeling to enrich MAIT cells. These sorted cells were then stimulated with 5-(2-oxopropylideneamino)-6-D-ribitylaminouracil (5-OP-RU; 50 nM), a major MR1 ligand that can be recognized by MAIT cells [25,26], and cultured in C10 medium supplemented with human IL-2 (20 ng/mL), IL-7 (10 ng/mL), and IL-15 (10 ng/mL) for 1–2 weeks. During this time, 5-OP-RU was added into the cell culture on day 5, 7, and 11 to further enrich MAIT cells. If needed, the resulting MAIT cells could be further purified using FACS or MACS via human TCR Vα7.2 antibody (BioLegend, San Diego, CA, USA) staining.

To generate iNKT cells, PBMCs were MACS-sorted via anti-iNKT MicroBead (Miltenyi Biotec) labeling to enrich iNKT cells, which were then stimulated with donor-matched irradiated αGC-PBMCs at the ratio of 1:1, and cultured in C10 medium supplemented with human IL-7 (10 ng/mL) and IL-15 (10 ng/mL) for 2–3 weeks. If needed, the resulting iNKT cells could be further purified using FACS via human iNKT TCR antibody (Clone 6B11; BD Biosciences) staining.

To generate γδT cells, PBMCs were stimulated with Zoledronate (5 mM; Sigma-Aldrich) and cultured in C10 medium supplemented with human IL-2 (20 ng/mL) for 2 weeks. If needed, the resulting PBMC-γδT cells could be further purified using FACS via human TCR Vδ2 antibody (Clone B6; BioLegend, San Diego, CA, USA) staining or via MACS using a human TCR γ/δ T Cell Isolation Kit (Miltenyi Biotec, Bergisch Gladbach, Germany).

To generate αβT cells, PBMCs were stimulated with CD3/CD28 T-activator beads (Thermo Fisher Scientific, Waltham, Massachusetts, USA) and cultured in C10 medium supplemented with human IL-2 (20 ng/mL; PeproTech) for 2–3 weeks, following the manufacturer’s instructions.

### 2.6. Generation of Human Mesothelin CAR-Engineered αβT (MCAR-αβT) and Mesothelin CAR-Engineered MAIT (MCAR-MAIT) cells

To generate MCAR-αβT cells, healthy donor PBMCs were cultured in a 12-well plate in C10 medium (1 × 10^6^ cells/mL/well) for 2 days, stimulated with CD3/CD28 T-activator beads (Thermo Fisher Scientific, Canoga Park, CA, USA) and supplemented with recombinant human IL-2 (20 ng/mL) (PeproTech). After 2 days, cells were spin-transduced with frozen-thawed Lenti/MCAR viruses supplemented with polybrene (10 μg/mL) at 660× *g* at 30 °C for 90 min following an established protocol [4]. Virus-transduced T cells were expanded for another 2–3 weeks in C10 medium with recombinant human IL-2 (20 ng/mL) (PeproTech) and cryopreserved for future use.

To generate MCAR-MAIT cells, PBMCs were MACS-sorted via PE-conjugated anti-TCR Vα7.2 antibody (BioLegend, San Diego, CA, USA) and anti-PE MicroBead (Miltenyi Biotec, Bergisch Gladbach, Germany) labeling to enrich MAIT cells. The sorted cells were stimulated with 5-OP-RU (50 nM) and cultured in C10 medium supplemented with human IL-2 (20 ng/mL), IL-7 (10 ng/mL), and IL-15 (10 ng/mL). After 2 days, cells were spin-transduced with freeze–thawed Lenti/MCAR viruses supplemented with polybrene (10 μg/mL) at 660× *g* at 30 °C for 90 min. Virus-transduced MAIT cells were expanded for another 1–2 weeks in C10 medium with recombinant human IL-2 (20 ng/mL), IL-7 (10 ng/mL), and IL-15 (10 ng/mL), and cryopreserved for future use. MCAR expression levels on αβT and MAIT cells were determined using flow cytometry. MCAR^+^ cell populations were FACS-sorted for the following assays.

### 2.7. In Vitro Tumor Cell Killing Assay

Tumor cells (1 × 10^4^ cells per well) were co-cultured with effector cells (at ratios indicated in figure legends) in C10 medium in Corning 96-well clear-bottom black plates. After 24 h, live tumor cells were quantified by adding D-luciferin (150 mg/mL; Caliper Life Science, Hopkinton, MA, USA) to cell cultures and reading out luciferase activities using an Infinite M1000 microplate reader (Tecan, Mannedorf, Switzerland).

### 2.8. In Vitro Mixed Lymphocyte Reaction (MLR) Assay

Healthy donor PBMCs were irradiated at 2500 rads and used as stimulators to study the graft-versus-host (GvH) response of conventional αβT and innate T cells as responders. Stimulators (5 × 10^5^ cells/well) and responders (2 × 10^4^ cells/well) were co-cultured in 96-well round-bottom plates in C10 medium for 4 days; the cell culture supernatants were then collected to measure IFN-γ production using ELISA.

### 2.9. In Vitro Mixed Mφ/T Reaction Assay

Healthy donor PBMC-derived M2-polarized macrophages were cocultured with donor-mismatched αβT, MAIT, iNKT, or γδT cells at a ratio of 1:1 in 96-well round-bottom plates in C10 medium for 24 h. At the end of cell culture, these cells were collected to study surface marker expression using flow cytometry, and the cell culture supernatants were collected to measure cytokine production using ELISA. In the mixed Mφ/MAIT reaction assay, 50 nM 5-OP-RU was added to co-cultures to activate MAIT cells; 10 μg/mL LEAF^TM^ purified anti-human MR1 (Clone 26.5, BioLegend, San Diego, CA, USA) or LEAF^TM^ purified mouse IgG2aκ isotype control antibody (Clone MOPC-173, BioLegend, San Diego, CA, USA) was added to co-cultures to study MAIT TCR-mediated target cell killing mechanism. In the mixed Mφ/iNKT reaction assay, 100 ng/mL αGC was added to co-cultures to activate iNKT cells; 10 μg/mL LEAF^TM^ purified anti-human CD1d (Clone 51.1, BioLegend, San Diego, CA, USA) or LEAF^TM^ purified mouse IgG2bκ isotype control antibody (Clone MG2b-57, BioLegend, San Diego, CA, USA) was added to co-cultures to study iNKT TCR-mediated target cell killing mechanism. In the mixed Mφ/γδT reaction assay, 5 μM zoledronate was added to co-cultures to activate γδT cells.

### 2.10. 3D Tumor/TAM/T-Cell Organoid Culture

The OVCAR3-FG tumor cells, M2-polarized macrophages, and MCAR-αβT or MCAR-MAIT cells were mixed at a ratio of 2:1:2 in C10 medium. Mixed cells were centrifuged, resuspended in C10 medium at 1 × 10^5^ cells per μL, adjusted to 5–10 μL per aggregate, and gently transferred onto a microporous membrane cell insert (Millicell, Billerica, MA, USA, #PICM0RG50) to form a 3D human tumor/TAM/T-cell organoid. Two days later, the organoids were dissociated to generate single-cell suspensions for further analysis.

### 2.11. Statistical Analysis

GraphPad Prism 6 (GraphPad software) was used for statistical data analysis. Student’s two-tailed *t* test was used for pairwise comparisons. Ordinary one-way ANOVA followed by Tukey’s or Dunnett’s multiple comparisons test was used for multiple comparisons. Data are presented as the mean ± SEM for more than three independent experiments, unless otherwise indicated. In all figures and figure legends, “n” represents the number of samples utilized in the indicated experiments. A p value of less than 0.05 was considered significant. ns, not significant; * *p* < 0.05; ** *p* < 0.01; *** *p* < 0.001; **** *p* < 0.0001.

## 3. Results

### 3.1. Generation and Characterization of Human Peripheral Blood Mononuclear Cell (PBMC)-Derived Innate T Cells

Human conventional αβ T cells and innate T cells, including mucosal-associated invariant T (MAIT), invariant natural killer T (iNKT), and gamma delta T (γδT) cells were generated from healthy donor PBMCs (Figure 1A,C,E,G). We routinely achieved over 95% purity of resulting T cells (Figure 1B,D,F,H). Conventional αβ T cells expanded ~500–1000 folds in 2–3 weeks, MAIT cells expanded~50–100 folds in 1–2 weeks, iNKT cells expanded 500–1000 folds in 2–3 weeks, and γδT cells expanded ~300–500 folds in ~2 weeks (data not shown). These T cells displayed different CD4/CD8 co-receptor expressions: conventional αβ T cells were mainly CD4^+^CD8^−^ (CD4 SP) or CD4^−^CD8^+^ (CD8 SP) cells (Figure 1B); MAIT cells were predominantly CD8 SP (70–90%), with some CD4^−^CD8^−^ (DN) (10–20%), and a minor population of CD4 SP cells (Figure 1D) [27,28]; iNKT cells contained three major populations of CD8 SP, CD4 SP, and DN (Figure 1F) [29]; γδT cells were mainly DN (70–90%), with some CD8 SP (10–20%), and very few CD4 SP (Figure 1H) [30].

To evaluate the antitumor capacity and TCR function of these innate T cells, we set up an in vitro tumor cell killing assay (Figure 2A,D,G). Two human tumor cell lines were used as targets, including a melanoma cell line A375 and a multiple myeloma cell line MM.1S. The two parental tumor cell lines were engineered to overexpress the firefly luciferase and enhanced GFP dual reporters (FG), and/or human CD1d, resulting in A375-FG, A375-CD1d-FG, MM.1S-FG, and MM.1S-CD1d-FG cell lines. Both A375 and MM.1S cells express MR1, which presents antigenic bacterial metabolites (e.g., 5-OP-RU) to MAIT TCR (Figure 2B) [28], and CD277, which plays a key role in phosphorylated metabolite (e.g., zoledronate)-induced activation of γδT cells (Figure 2H) [31]. The engineered A375-CD1d-FG and MM.1S-CD1d-FG cells express CD1d, which presents glycolipids (e.g., αGC) to iNKT TCR and activates iNKT cells (Figure 2E and Appendix A) [32]. In the presence of 5-OP-RU, αGC, or zoledronate, MAIT, iNKT, and γδT cells killed tumor cells with an improved efficacy (Figure 2C,F,I), confirming the potent antitumor capacity of these innate T cells mediated by their TCR recognition. Certain levels of tumor cell killing were observed in the absence of TCR agonists (Figure 2F,I), indicating the intrinsic NK activating receptor (e.g., NKG2D, NKp30, and NKp44)-mediated tumor cell killing of innate T cells [4,8,28,33,34].

Graft-versus-host (GvH) response is the foremost safety concern of off-the-shelf allogeneic cellular products [1,4]. MAIT, iNKT, and γδT cells do not recognize mismatched human leukocyte antigen (HLA) molecules and protein alloantigens; as a result, these innate T cells are considered not to induce GvH responses [4,7,8,35,36,37,38]. To verify the GvH-free feature of the innate T cells, we performed an in vitro mixed lymphocyte reaction (MLR) assay (Figure 2G). Different from conventional αβ T cells, innate T cells, including MAIT, iNKT, and γδT cells, did not react to the mismatched healthy donor PBMCs, evidenced by the lack of IFN-γ secretion (Figure 2K). Therefore, the GvH-free feature of innate T cells grant them a high safety profile and make them suitable for off-the-shelf allogeneic cell therapy.

### 3.2. Targeting Immunosuppressive M2-Polarized Macrophages by MAIT Cells

To induce M2 polarization, human PBMCs were first cultured with M-CSF for 6 days and subsequently differentiated into M2 polarized macrophages by adding the anti-inflammatory stimuli IL-4 and IL-13 for another 2 days (Figure 3A,B). Indeed, after polarization, we observed upregulation of M2 macrophage markers CD163 and CD206 (Figure 3A–C) [39,40,41].

To study the macrophage-targeting capacity of innate T cells, we set up an in vitro mixed Mφ/T reaction assay (Figure 3D,F). M2 macrophages that upregulated MR1 could be recognized by MAIT TCR (Figure 3G). Unlike conventional αβ T cells, which did not target M2 macrophages (Figure 3E), MAIT cells killed M2 macrophages effectively in the presence of 5-OP-RU, and the killing capacity was dampened by blocking MR1, attesting to the macrophage-targeting potency of MAIT cells in an MR1-antigen-MAIT TCR recognition (Figure 2H) [42]. Interestingly, even without the addition of 5-OP-RU, MAIT cells could still effectively kill M2 macrophages, which may account for their intrinsic NK activating receptor-mediated function (Figure 2H) [43]. The macrophage-targeting and killing by MAIT cells were correlated with the upregulation of activation markers (i.e., CD25; Figure 3I,J) and generation of pro-inflammatory cytokines (i.e., IFN-γ; Figure 3K). In the mixed Mφ/MAIT assay, the residual M2 macrophages, after co-culturing with MAIT cells, displayed a significant reduction in MR1 expression level (Appendix A). One possible reason was that anti-inflammatory stimuli (i.e., IL-4 and IL-13) induced macrophage polarization and upregulation of MR1, and the MR1^high^ macrophages were more easily recognized and killed by MAIT cells, resulting in a residual MR1^low^ macrophage population (Appendix A). Collectively, MAIT cells can potentially limit the macrophage-modulated immunosuppression by targeting and killing macrophages.

### 3.3. Targeting Immunosuppressive M2-Polarized Macrophages by iNKT Cells

One attractive feature of iNKT cells is that iNKT cells can alter the solid tumor immunosuppressive TME via inhibition of immunosuppressive TAMs and myeloid-derived suppressive cells (MDSCs); these cells express CD1d, and thereby can be recognized by iNKT cells (Figure 4B) [24,44,45,46]. In an in vitro mixed Mφ/iNKT reaction assay, iNKT cells effectively killed M2 macrophages in the absence of αGC by virtue of their NK activating receptor-mediated function (Figure 4A,C). The addition of αGC significantly enhanced iNKT cell-induced killing of macrophages, which could be blocked using the anti-CD1d antibody (Figure 4C), validating CD1d-antigen-iNKT TCR-mediated macrophage recognition and killing by iNKT cells. In addition, macrophage killing by iNKT cells was correlated with their enhanced cytokine secretion and activation marker expression (Figure 4D–F). Similar to the residual M2 macrophages in the mixed Mφ/MAIT assay (Appendix A), these cells, after co-culturing with iNKT cells, also displayed a significant reduction in CD1d expression level (Figure 4G), indicating a prior targeting of CD1d^high^ macrophage population by iNKT cells.

### 3.4. Targeting Immunosuppressive M2-Polarized Macrophages by γδT Cells

The functions of γδT cells in TME modulation are controversial. In recent years, there have been a number of ongoing reports claiming that γδT cells can raise the population of myeloid-derived suppressor cells (MDSCs) and facilitate cancer progression [47,48,49], while other studies demonstrate that zoledronic acid can induce powerful γδT cell-mediated antitumor responses, and trigger γδT cells to target monocytes and downregulate inflammatory homing [50]. Here, we utilized an in vitro Mφ/γδT cell assay to study the killing of M2 macrophages, wherein γδT cells were co-cultured with M2 macrophages with or without the addition of zoledronate (Figure 5A). Similar to MAIT and iNKT cells, γδT cells by themselves demonstrated a strong killing capacity towards macrophages, which may have been a result of their NK function [51,52]. This macrophage killing was further enhanced by the addition of zoledronate (Figure 5B) [31,53]. The enhanced killing capacity was correlated with the increased CD25 expression and IFN-γ secretion of γδT cells (Figure 5C–E). Previous studies showed the generation of activating or inhibitory anti-CD277 monoclonal antibodies, which can induce or inhibit γδT cell activation and proliferation, respectively [54,55]. These antibodies can also have similar activating or suppressive effects on the γδT cells’ macrophage-targeting capacity.

### 3.5. Targeting Tumor-Associated Macrophages (TAMs) by Mesothelin-Targeting CAR (MCAR)-Engineered MAIT (MCAR-MAIT) Cells

To mimic TME and study the immunosuppressive function of TAMs, we set up a 3D tumor/TAM/T-cell organoid culture [19]. Mesothelin (MSLN) was used as the model tumor antigen. MSLN-targeting CAR-αβT (MCAR-αβT), and MSLN-targeting CAR-MAIT (MCAR-MAIT) cells were generated by transducing healthy donor PBMCs with a Lenti/MCAR lentivector and sorting for CAR^+^ populations (Figure 6A–C and Appendix A). A human ovarian cancer cell line OVCAR3-FG overexpressing MSLN and Fluc-GFP was used as a target (Figure 6E). As a proof-of-principle, here we chose MCAR-MAIT cells as an innate T cell type to study the interactions between tumor cells, TAMs, and immune cells. MACR-engineered iNKT and γδT cells would also be very interesting to study in the future. Both MCAR-αβT and MCAR-MAIT cells effectively killed OVCAR3-FG tumor cells, and 5-OP-RU could further enhance the tumor cell killing capacity of MCAR-MAIT cells, indicating their CAR/TCR dual mechanisms for targeting OVCAR3 (Figure 6D,F). In the 3D tumor organoid culture (Figure 6G), M2-polarized macrophages suppressed MCAR-αβT-mediated killing of tumor cells (Figure 6H). Accordingly, MCAR-αβT co-cultured with M2-polarized macrophages, compared to those that were not co-cultured with macrophages, showed decreased expression of the T cell activation marker (i.e., CD25; Figure 6I). However, MCAR-MAIT cells sustained their potent antitumor capacity in the presence of macrophages, evidenced by the comparable or even increased OVCAR3 tumor cell killing (Figure 6K). In addition, unlike MCAR-αβT cells, MCAR-MAIT cells displayed a sustained expression of T cell activation marker CD25 (Figure 6L). The targeting and killing of TAMs by MCAR-MAIT cells may account for their sustained tumor killing capacity and activation (Figure 6J,M, and Appendix A). Collectively, these results support a cancer therapy potential of human CAR-MAIT cells, which can target TAMs in TME and maintain their antitumor activity.

## 4. Discussion

TAMs have been demonstrated to promote tumor angiogenesis and tumor cell survival [16]. A variety of studies showed that depleting TAMs or inhibiting their functions could limit tumor progression, making tumor cells targets for cancer immunotherapy. Methods include (1) reducing the TAM population using bisphosphonates and their derivatives [56], or targeting CSF-1/CSF-1R signaling [57]; (2) reverting M2 TAMs to an M1-like phenotype by inducing pro-inflammatory cytokines [58], or intervening with specific receptors such as TLRs [59]; (3) regulating the macrophage phagocytosis signal by targeting CD47-SIRP-α, MHC-1/LILRB1, or CD24/Siglec-10 signaling [60,61,62]; (4) triggering macrophage phagocytosis by engineering CARs on macrophages [63,64]. Therefore, combining current immunotherapy techniques, such as checkpoint inhibitor blockade and CAR-T cell therapy, with TAM-targeting drugs will significantly improve the specificity and efficacy of these cancer treatments.

This study successfully established CAR-engineered innate T cells as both antitumor effector cells and remediators of TME immunosuppression through the targeting of anti-inflammatory TAMs. Compared to the conventional CAR-T cell therapy, which should be coupled with TAM-targeting drugs, the CAR-engineered innate T cell therapy itself is able to effectively target both tumor cells and TAMs, improving antitumor efficacy and reducing the potential adverse side effects induced by drugs. All three innate T cell types, including MAIT, iNKT, and γδT cells, exhibited potent Mφ-killing capacity (Figure 3, Figure 4 and Figure 5), and these innate T cells could be engineered to express tumor-targeting CARs that maintain this function in a 3D organoid culture mimicking the TME (Figure 6). This development could be incredibly beneficial for the advancement of the field of CAR therapy for several key reasons. As innate immune cells, the therapies examined in this paper do not have the same proclivity for GvH responses as conventional αβ T cell-based CAR therapy, making them much more promising as candidates for universal “off-the-shelf” allogeneic therapy [3,5,65]. This would greatly expand the scalability of antitumor CAR therapy, making it much more accessible to patients with limited resources and to countries with poor or unevenly distributed medical infrastructure, as well as patients with truncated time scales that cannot accommodate the time limitations of autologous CAR-T generation [66]. Additionally, the Mφ-killing abilities established by this study provide a multifaceted approach to tumor killing, expanding the scope of CAR therapy from solely antitumor cytotoxicity to the re-engineering of the TME towards a less anti-inflammatory state [67].

These results are incredibly promising for the future of innate immune cell-based CAR therapy; however, there is much work that needs to be done before they can be applied at a large scale. The low numbers and high variabilities of these innate T cells in cancer patients, as well as the limited in vitro expansion of these cells, are considered the major factors limiting innate T cell-based translational and clinical applications. These innate T cells were tested exclusively in vitro, and require in vivo replication before their TAM-targeting nature can be fully verified. While organoid culture can mimic the TME, animal models and clinical replication are necessary to characterize the full, on-site efficacy of CAR-innate immune cells as both antitumor effector cells and TAM-targeting cells. Of note, in the TME, M1 macrophages have been shown to suppress tumor growth by engulfing the target tumor cells and through antibody-dependent cell-mediated cytotoxicity (ADCC). The proposed CAR-engineered innate T cell therapy could also potentially target M1 macrophages in the TME and limit their antitumor reactivity [68]. Further explorations on the M1 macrophage-targeting capacity of innate T cells and the corresponding changes in the antitumor function are necessary. Additionally, this study only demonstrated Mφ- and tumor-killing abilities of MCAR-MAIT cells in an organoid culture. While there have been many studies on CAR-iNKT and γδT cells, more representative studies are needed to verify their improved antitumor efficacy as a result of Mφ depletion [4,69].

## 5. Conclusions

Overall, these results suggest an impressive new pathway via which CAR therapy can target tumors. The efficacy of MAIT, iNKT, and γδT cells in targeting tumor-associated macrophages in vitro lays the groundwork for future studies to expand the scope of antitumor CAR therapy and target one of the main defenses of tumors against immune infiltration. This is a promising development for the expansion of CAR therapy into solid tumors, and could improve both the efficacy and accessibility of one of the most exciting fields in cancer immunotherapy.

## Figures and Tables

**Figure 1 cancers-14-02749-f001:**
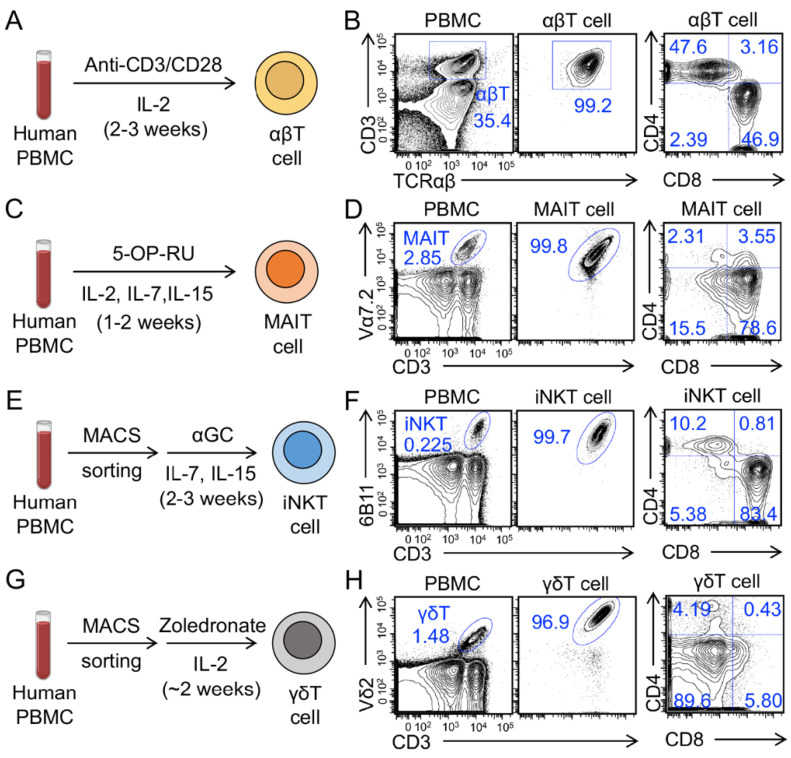
Generation of human peripheral blood mononuclear cell (PBMC)-derived T cells. (**A**,**C**,**E**,**G**) Diagram outlining the generation of PBMC-derived conventional αβ T (αβT), mucosal-associated invariant T (MAIT), invariant natural killer T (iNKT), and gamma delta T (γδT) cells. (**B**,**D**,**F**,**H**) Fluorescence-activated cell sorting (FACS) detection of the generation and CD4/CD8 co-receptor expression of αβT, MAIT, iNKT, and γδT cells. Representative of over five experiments.

**Figure 2 cancers-14-02749-f002:**
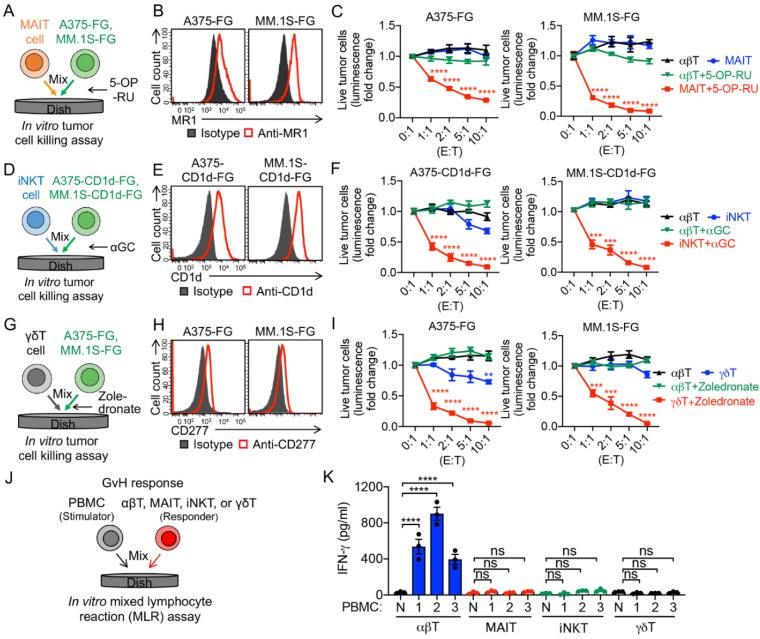
In vitro antitumor tumor efficacy and safety study of PBMC-derived T cells. (**A**–**C**) In vitro killing of human A375 melanoma and MM.1S multiple myeloma cells by MAIT cells. Both tumor cell lines were engineered to express firefly luciferase and green florescence protein (FG) dual reporters. Moreover, 5-(2-oxopropylideneamino)-6-d-ribitylaminouracil (5-OP-RU) was added to activate MAIT cells. (**A**) Experimental design. (**B**) FACS detection of MR1 on A375-FG and MM.1S-FG cells. (**C**) Tumor killing data at 24 h (n = 4). (**D**–**F**) In vitro killing of A375 and MM.1S cells by iNKT cells. Both tumor cell lines were engineered to express human CD1d as well as FG dual reporters, and α-galactosylceramide (αGC) was added to activate iNKT cells. (**D**) Experimental design. FACS detection of CD1d on A375-CD1d-FG and MM.1S-CD1d-FG cells. (**F**) Tumor killing data at 24 h (n = 4). (**G**–**I**) In vitro killing of A375 and MM.1S cells by γδT cells. Zoledronate was added to activate γδT cells. (**G**) Experimental design. (**H**) FACS detection of CD277 on A375-FG and MM.1S- FG cells. (**I**) Tumor killing data at 24 h (n = 4). (**J**,**K**) Studying the graft-versus-host (GvH) response of PBMC-derived T cells using an in vitro mixed lymphocyte reaction (MLR) assay. (**J**) Experimental design. PBMCs from three different healthy donors were used as stimulator cells. (**K**) ELISA analyses of IFN-γ secretion at day 4 (n = 3). N, no stimulator cells. Representative of three experiments. Data are presented as the mean ± SEM. ns, not significant, ** *p* < 0.01, *** *p* < 0.001, **** *p* < 0.0001, by one-way ANOVA.

**Figure 3 cancers-14-02749-f003:**
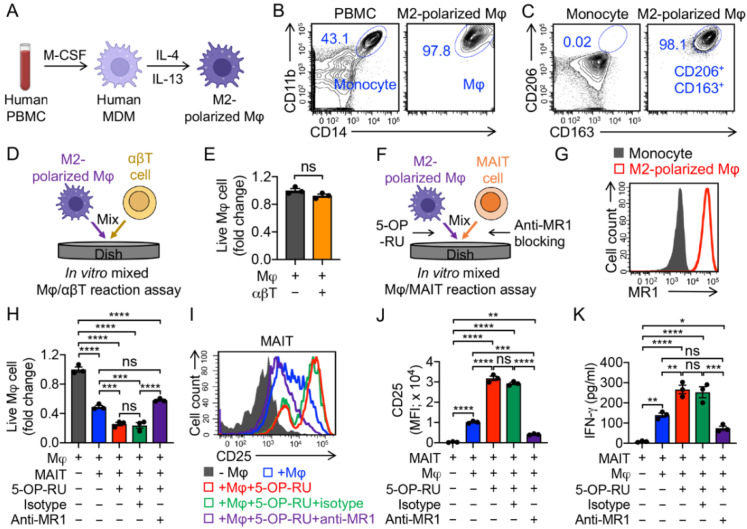
In vitro targeting of immunosuppressive macrophages by PBMC-derived MAIT cells. (**A**–**C**) In vitro generation and polarization of human monocyte-derived M2 macrophages. (**A**) Experimental design. M-CSF, macrophage colony-stimulating factor; MDM, monocyte-derived macrophage; Mφ, macrophage. (**B**) FACS detection of CD11b and CD14 on M2-polarized macrophages. Healthy donor PBMCs were included as a staining control. (**C**) FACS detection of M2 macrophage markers (i.e., CD163 and CD206) on M2-polarized macrophages. Monocytes were included as a control. (**D**,**E**) Studying macrophage targeting by αβ T cells using an in vitro mixed macrophage/αβ T cell (Mφ/αβT) reaction assay. (**D**) Experimental design. (**E**) FACS analysis of live macrophages 24 h after co-culturing with αβ T cells. Live cells were identified as e506^−^CD14^+^CD11b^+^ (n = 3). (**F**–**K**) Studying macrophage targeting by MAIT cells using an in vitro mixed macrophage/MAIT cell (Mφ/MAIT) reaction assay; 5-OP-RU was added to activate MAIT cells. (**F**) Experimental design. (**G**) FACS detection of MR1 on monocytes and M2-polarized macrophages. (**H**) FACS analysis of live macrophages 24 h after co-culturing with MAIT cells. (**I**) FACS detection of CD25 expression on MAIT cells. (**J**) Quantification of I (n = 3). (**K**) ELISA analysis of IFN-γ secretion by MAIT cells in the supernatants of various mixed cell cultures (n = 3). Representative of three experiments. Data are presented as the mean ± SEM. ns, not significant, * *p* < 0.05, ** *p* < 0.01, *** *p* < 0.001, **** *p* < 0.0001, by Student’s *t* test (**E**), or by one-way ANOVA (**H**,**J**,**K**).

**Figure 4 cancers-14-02749-f004:**
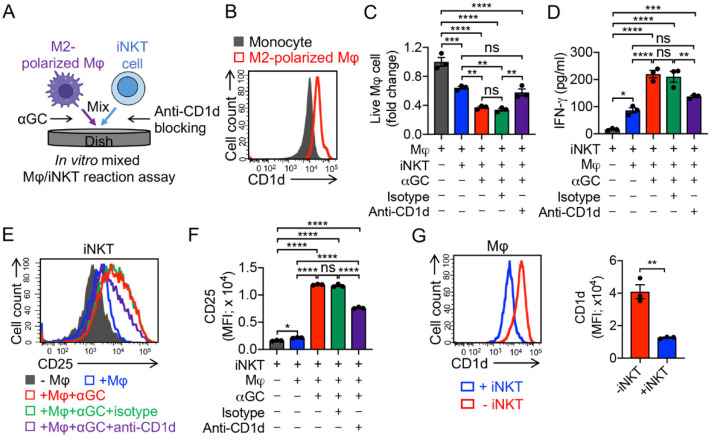
In vitro targeting of immunosuppressive macrophages by PBMC-derived iNKT cells. (**A**) Experimental design. αGC was added to activate iNKT cells. (**B**) FACS detection of CD1d on monocytes and M2-polarized macrophages. (**C**) FACS analysis of live macrophages 24 h after co-culturing with iNKT cells. (**D**) ELISA analysis of IFN-γ secretion by iNKT cells in the supernatants of various mixed cell cultures (n = 3). (**E**) FACS detection of CD25 expression on iNKT cells. (**F**) Quantification of E (n = 3). (**G**) FACS analysis of CD1d expression on macrophages with or without co-culturing with iNKT cells. Representative of three experiments. Data are presented as the mean ± SEM. ns, not significant, * *p* < 0.05, ** *p* < 0.01, *** *p* < 0.001, **** *p* < 0.0001, by Student’s *t* test (**G**), or by one-way ANOVA (**C**,**D**,**F**).

**Figure 5 cancers-14-02749-f005:**
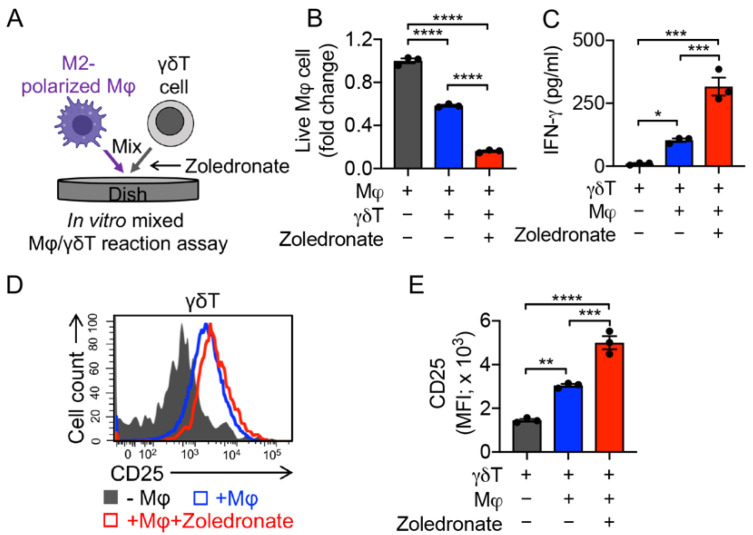
In vitro targeting of immunosuppressive macrophages by PBMC-derived γδT cells. (**A**) Experimental design. Zoledronate was added to activate γδT cells. (**B**) FACS analysis of live macrophages 24 h after co-culturing with γδT cells. (**C**) ELISA analysis of IFN-γ secretion by γδT cells in the supernatants of various mixed cell cultures (n = 3). (**D**) FACS detection of CD25 expression on γδT cells. (**E**) Quantification of D (n = 3). Representative of three experiments. Data are presented as the mean ± SEM. ns, not significant, * *p* < 0.05, ** *p* < 0.01, *** *p* < 0.001, **** *p* < 0.0001, by one-way ANOVA.

**Figure 6 cancers-14-02749-f006:**
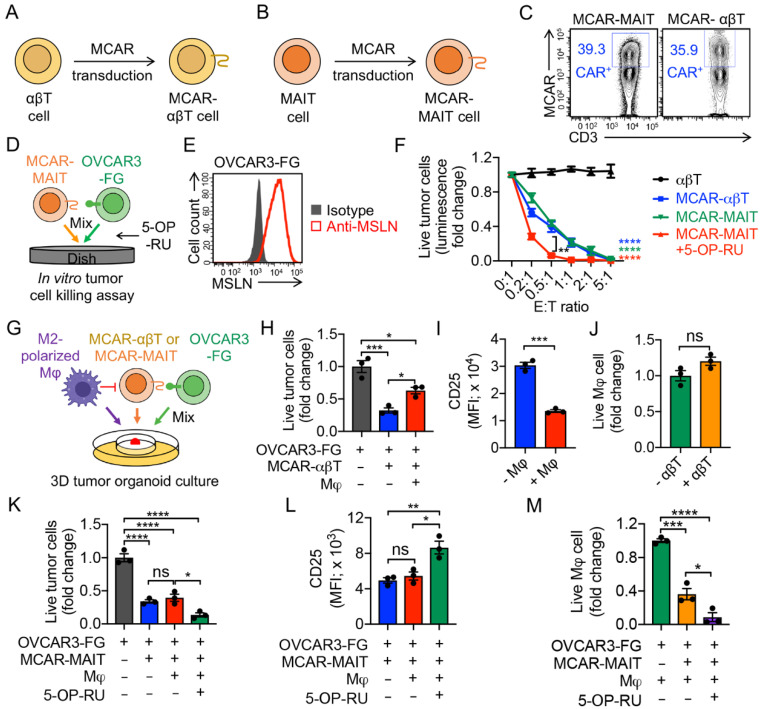
Targeting TAMs by mesothelin-targeting CAR-engineered MAIT (MCAR-MAIT) cells using an ex vivo 3D TME mimicry culture. (**A**,**B**) Diagram outlining the generation of MCAR-engineered αβT (MCAR-αβT) and MAIT cells. (**C**) FACS detection of MCAR expression on MCAR-αβT and MCAR-MAIT cells. (**D**–**F**) In vitro killing of human ovarian cancer OVCAR3 cells by MCAR-MAIT cells. OVCAR3-FG, human OVCAR3 cell line engineered to overexpress FG. MCAR-αβT cells were included as an effector cell control. MCAR-MAIT and MCAR-αβT cells were FACS-sorted for MCAR^+^ cell populations. (**D**) Experimental design. (**E**) FACS detection of mesothelin (MSLN) expression on OVCAR3-FG cells. (**F**) Tumor killing data at 24 h (n = 4). (**G**–**M**) Studying TAM targeting by MCAR-MAIT cells using an ex vivo 3D TME mimicry culture. MCAR-αβT cells were included as an effector cell control. (**G**) Experimental design. (**H**) FACS analysis of live OVCAR3-FG cells 48 h after co-culturing with MCAR-αβT cells. (n = 3). (**I**) FACS detection of CD25 expression on MCAR-αβT cells (n = 3). (**J**) FACS analysis of live macrophages 48 h after co-culturing with MCAR-αβT cells (n = 3). Live macrophages were identified as e506^−^CD14^+^CD11b^+^. (**K**) FACS analysis of live OVCAR3-FG cells 48 h after co-culturing with MCAR-MAIT cells. (n = 3). (**L**) FACS detection of CD25 expression on MCAR-MAIT cells (n = 3). (**M**) FACS analysis of live macrophages 48 h after co-culturing with MCAR-MAIT cells (n = 3). Representative of three experiments. Data are presented as the mean ± SEM. ns, not significant, * *p* < 0.05, ** *p* < 0.01, *** *p* < 0.001, **** *p* < 0.0001, by Student’s *t* test (**F**,**I**,**J**), or by one-way ANOVA (**H**,**K**–**M**).

## Data Availability

Data supporting reported results are available on request from the corresponding author.

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
