# Peer review of "Targeting Immunosuppressive Tumor-Associated Macrophages Using Innate T Cells for Enhanced Antitumor Reactivity"

_cancers, 2022, doi:10.3390/cancers14112749_

Round 1
Reviewer 1 Report
In this manuscript by Ji et al, authors employ different innate-(like) cells to further expand the toolbox for cellular therapies. The results are displayed in a pleasing fashion, and most experiments are performed with a proper experimental design / proper controls, with convincing data. However, the seminal figure with CAR+ cells is not very convincing, or fair. Please see my comments below how authors could fix this – hopefully, I missed the details on how these cells were FACS-sorted before inclusion into the killing experiments. Otherwise, these experiments need to be repeated before they can be included in the manuscript. Please also find my other recommendations below.
Major
- I’m not really happy with the CD86 staining histograms shown throughout the manuscripts – the PMT voltage was probably set too high, as for some conditions, the histogram goes off of the scale at the right side, and that’s not good. Authors should supply either an isotype staining histogram, showing that the background staining is normal (i.e. between 0-10^3), or provide representative histograms after establishing correct PMT values for the channel CD86 was measured in. Normally I wouldn’t mind, but authors make a whole statement about which cells are targeted and which aren’t. Alternatively, these panels and the statements regarding CD86-expressing cells can be removed from the manuscript.
- As mentioned above, I do not agree with the experimental setup of figure 6. If I read the materials and methods and figure legend and look at the instructionary panels, it is not clear to me whether these MCAR+ cells were FACS-sorted out or not and/or whether the authors compensated the “effector ratio” dependent on the amount of MCAR+ cells if no sorting was employed. In panel C, it seems like transduction efficiencies are about equal, which is fine. However, if one wants to compare ab T cells with MAIT T cells, then the innate-killing capacity of the MAIT T cells (shown in Figure 3 by the authors) should be corrected for. It is not fair to add 65-ish% of cells that can kill the target cells and then say “MAIT cells are better”. Hopefully, authors did employ FACS-sorting to obtain pure populations before starting their co-cultures/3D models and just did not add this to the materials and methods. If so, please add, otherwise, these experiments need to be repeated with FACS-sorted cells to really be sure if the effects observed with the MAIT cells are not induced by natural killing capacity of these cells, rather than the transduced mesothelin-specific CAR.
Minor
General
- Please add the log axis to all flow cytometry plots so readers can see height of expression/manipulation of axes. For instance, in figure 1, there seem to be substantial parts of the populations on the borders of the dot plots, which often means that the axes are cut differently than expected. This is not per se a bad thing, but having the log-axes there can help for understanding.
- Please provide a rationale in the main text why MCAR-iNKT and MCAR-yd T cells were not included in the last part of the study.
Line 114 – please add dilutions and employed fluorochromes of all antibodies employed in this study (i.e. in a supplementary table).
Line 128 – please add FlowJo software version.
Line 193 – spin-transduced is more common I believe, suggest to change.
Line 195 – no additional TCR stimulation was performed after 2 weeks of culture?
Line 206 – there’s a double period.
Figure 1 – maybe it could be nice to add the culture period time under the arrows with the cytokine cocktail? Just a suggestion. Furthermore, it would be great of authors could provide data on cellular expansion of the different cell types. This could then also be reflected on in the discussion, as probably, quite a large quantity of cells is required for treatment, and this might be a limitation (or not, if these cells expand tremendously).
Figure 2 – for E, to be convincing, please add staining in parental cells. The shift in MFI is marginal (probably as expected in these cell lines with these difficult to stain for markers), so it would be good if authors could add data from the parental cell lines, which should overlap with the isotype peak.
Line 286, 349 – aggressively is not really appropriate, please delete.
Line 312 – there’s a double comma.
Line 327 – I don’t like the mixing of literature and results the way performed here; please pull apart the introductionary paragraph and description of the results. Adding something like “To induce M2 polarization, human PBMCs were first cultured with M-CSF and subsequently differentiated into M2 polarized macrophages by culturing with the anti-inflammatory stimuli IL-4 and IL-13 (Figure 3A, B). Indeed, after X days, we observed upregulation of macrophage activation markers CD163 and CD206 (Figure 3C).” - this might seem a bit excessive, but now, for a glancing reader, it seems like you also tested IL-10 production and PD-L1 expression, which you didn’t (which again is not a problem, but it’s better to make the distinction more clear).
Line 406 – previous studies would be better.
Line 435 – how is CD25 upregulation equivalent to or indicative of sustained antitumor capacity? Sensing IL-2 does not necessarily make a better killer. Please rephrase this sentence.
Line 471 – please change temporal to time, works just fine and some people might confuse temporal for temporary.
Reviewer 2 Report
This interesting and important paper can be published as it stands.
Author Response
We appreciate the Reviewer's comments.
Reviewer 3 Report
The manuscript deals with a very important and novel topic, with potential clinical relevance. However, the in vitro models represent a limitation for clinical immediate translation, as the authors also indicated in the discussion. Notably, the introduction presents some confusing and uncorrect concept that should be addressed. The Discussion is very scarce and need significant improvement
Specific major comments:
- It is not clear for the authors what the authors mean for innate T cells. Please describe the parameters that define these peculiar cells.
- lines 48-49, and lines 55-56: This concept is not clear and univocal. Indeed, innate immune response include also macrophagic response and related inflammation. Moreover, immune checkpoint activation is intrinsic to T cell antigen mediated activation, and it is not only a disease-related defence mechanism against host immunity. These sentences are at least incomplete and partially uncorrect.
- Lines 55-56: Macrophage may polarize toward a proinflammatory M1 profile or anti-inflammatory M2 profile. This may depend by various factors and mediators, including tumor type and tumor microenvironment conditions. Moreover, also inflammation and proinflammatory TAM results in immunosuppression. See at this regard explanatory reviews in literature (as Immunology. 2020;159(4):357-364, and others).
- The Discussion is too short and scarce. Please comment the potential application and role of inhibiting M2 polarization in TME and the implication of change TME from an anti-inflammatory to a less anti.-inflammatory or pro-inflammatory state. What could be in ex-vivo and human TME the role of the presence of a mixed TAM population including usually both M1 and M2 TAMs?
Reviewer 4 Report
cancers-1716133-peer-review-v1
General comments: This manuscript communicates a well-designed study – of which the use of multiple cell lines is commended. Excellent figures, appropriate statistics, and very well written. No changes or edits requested.
Author Response
We appreciate the Reviewer's comments.
Round 2
Reviewer 1 Report
Dear authors, thank you for taking my suggestions into consideration. I have a minor point that can be corrected during proofreading - it's fold-change (with or without the -), not folds change when talking about expansion. I would like to congratulate the authors on a nice paper, I am going to recommend acceptation to the editors.